# The IBEX knowledge-base a community resource enabling adoption and development of immunofluorescence imaging methods

Ziv Yaniv[1]*, Ifeanyichukwu U Anidi[2], Leanne Arakkal[3], Armando J Arroyo-Mejías[3], Rebecca T Beuschel[3], Katy Börner[4], Colin J Chu[5], Beatrice H Clark[3], Menna R Clatworthy[6], Jake Colautti[7], Fabian Coscia[8], Joshua Croteau[9], Saven Denha[7], Rose Dever[10], Walderez O Dutra[11], Sonja Fritzsche[8,12], Spencer Fullam[13], Michael Y Gerner[14], Anita Gola[15], Kenneth J Gollob[16], Jonathan M Hernandez[17], Jyh Liang Hor[3], Hiroshi Ichise[3], Zhixin Jing[3], Danny Jonigk[18,19], Evelyn Kandov[3], Wolfgang Kastenmüller[20], Joshua FE Koenig[7], Aanandita Kothurkar[5], Rosa K Kortekaas[21], Alexandra Y Kreins[22], Ian T Lamborn[3], Yuri Lin[17], Katia Luciano Pereira Morais[16], Aleksandra Lunich[2], Jean CS Luz[23], Ryan B MacDonald[5], Chen Makranz[24], Vivien I Maltez[25], John E McDonough[21], Ryan V Moriarty[26], Juan M Ocampo-Godinez[22,27], Vitoria Murakami Olyntho[7], Annette Oxenius[28], Kartika Padhan[3,29], Kirsten Remmert[17], Nathan Richoz[6], Edward C Schrom[3], Wanjing Shang[3], Lihong Shi[30], Rochelle M Shih[3], Emily Speranza[31], Salome Stierli[32], Sarah A Teichmann[33], Tibor Z Verse[3], Megan Vierhout[7,21], Brianna T Wachter[34], Adam K Wade-Vallance[3], Margaret Williams[2], Nathan Zangger[28], Ronald N Germain[3,29], Andrea J Radtke[3,29,35]*

*For correspondence:
zivyaniv@nih.gov (ZY);
Andrea.Radtke@leica-microsystems.com (AJR)

[1]Bioinformatics and Computational Bioscience Branch, National Institute of Allergy and Infectious Diseases, National Institutes of Health, Bethesda, United States; [2]Critical Care Medicine and Pulmonary Branch, National Heart, Lung and Blood Institute, National Institutes of Health, Bethesda, United States; [3]Lymphocyte Biology Section, Laboratory of Immune System Biology, NIAID, NIH, Bethesda, United States; [4]Department of Intelligent Systems Engineering, Indiana University, Bloomington, United States; [5]UCL Institute of Ophthalmology and NIHR Moorfields Biomedical Research Centre, London, United Kingdom; [6]Cambridge Institute for Therapeutic Immunology and Infectious Diseases, University of Cambridge Department of Medicine, Molecular Immunity Unit, Laboratory of Molecular Biology, Cambridge, United Kingdom; [7]McMaster Immunology Research Centre, Schroeder Allergy and Immunology Research Institute, Department of Medicine, Faculty of Health Sciences, McMaster University, Hamilton, Canada; [8]Max-Delbrueck-Center for Molecular Medicine in the Helmholtz Association (MDC), Spatial Proteomics Group, Berlin, Germany; [9]Department of Business Development, BioLegend Inc, San Diego, United States; [10]Functional Immunogenomics Unit, National Institute of Arthritis and Musculoskeletal and Skin Diseases, National Institutes of Health, Bethesda, United States; [11]Laboratory of Cell-Cell Interactions, Department of Morphology, Institute of Biological Sciences, Universidade Federal de Minas Gerais, Belo Horizonte, Brazil; [12]Humboldt-Universität zu Berlin, Institute of Biology, Berlin, Germany; [13]Division of Rheumatology, Rush University Medical Center, Chicago, United States; [14]Department of Immunology, University of Washington School of Medicine, Seattle,

United States; [15]Robin Chemers Neustein Laboratory of Mammalian Cell Biology and Development, The Rockefeller University, New York, United States; [16]Center for Research in Immuno-oncology (CRIO), Hospital Israelita Albert Einstein, Sao Paulo, Brazil; [17]Surgical Oncology Program, National Cancer Institute, National Institutes of Health, Bethesda, United States; [18]Institute of Pathology, Aachen Medical University, RWTH Aachen, Aachen, Germany; [19]German Center for Lung Research (DZL), Biomedical Research in Endstage and Obstructive Lung Disease Hannover (BREATH), Hannover, Germany; [20]Würzburg Institute of Systems Immunology, Max Planck Research Group at the Julius-Maximilians-Universität Würzburg, Würzburg, Germany; [21]Department of Medicine, McMaster University, Firestone Institute for Respiratory Health, St Joseph's Healthcare, Hamilton, Canada; [22]Infection Immunity and Inflammation Research and Teaching Department, University College London Great Ormond Street Institute of Child Health, London, United Kingdom; [23]Viral Vector Laboratory, Cancer Institute of São Paulo, University of São Paulo, Sao Paulo, Brazil; [24]Neuro-Oncology Branch, National Cancer Institute, National Institutes of Health, Bethesda, United States; [25]Division of Allergy, Immunology and Rheumatology, Department of Pediatrics, University of California San Diego, La Jolla, United States; [26]Department of Cellular and Developmental Biology, Northwestern University, Chicago, United States; [27]Laboratorio de Bioingeniería de Tejidos, Departamento de Estudios de Posgrado e Investigación, Universidad Nacional Autónoma de México, Mexico City, Mexico; [28]Institute of Microbiology, ETH Zurich, Zurich, Switzerland; [29]Center for Advanced Tissue Imaging Laboratory of Immune System Biology, NIAID, NIH, Bethesda, United States; [30]Laboratory of Immune System Biology, National Institute of Allergy and Infectious Diseases, National Institutes of Health, Bethesda, United States; [31]Florida Research and Innovation Center, Cleveland Clinic Foundation, Port Saint Lucie, United States; [32]Institute of Anatomy, University of Zurich, Zurich, Switzerland; [33]Cambridge Stem Cell Institute, Jeffrey Cheah Biomedical Centre, Puddicombe Way,Cambridge Biomedical Campus, Cambridge, United Kingdom; [34]Laboratory of Clinical Immunology and Microbiology, National Institute of Allergy and Infectious Diseases, National Institutes of Health, Bethesda, United States; [35]Leica Microsystems, Wetzlar, Germany

## eLife Assessment

The IBEX Knowledge-Base is a **fundamental** tool that will enhance scientific collaboration by providing a centralized, community-driven resource for immunofluorescence imaging and reagent validation. Its detailed use cases, open-source design, and transparent reporting offer **exceptional** evidence of its broad utility and impact in the life sciences. It is now up to the community to contribute to its growth. Overall, the resource sets a high standard as a blueprint for future community initiatives in reproducibility and standardization.

**Abstract** The iterative bleaching extends multiplexity (IBEX) Knowledge-Base is a central portal for researchers adopting IBEX and related 2D and 3D immunofluorescence imaging methods. The design of the Knowledge-Base is modeled after efforts in the open-source software community and includes three facets: a development platform (GitHub), static website, and service for data archiving. The Knowledge-Base facilitates the practice of open science throughout the research life cycle by providing validation data for recommended and non-recommended reagents, such as primary and secondary antibodies. In addition to reporting negative data, the Knowledge-Base empowers method adoption and evolution by providing a venue for sharing protocols, videos, datasets, software, and publications. A dedicated discussion forum fosters a sense of community among

researchers while addressing questions not covered in published manuscripts. Together, scientists from around the world are advancing scientific discovery at a faster pace, reducing wasted time and effort, and instilling greater confidence in the resulting data.

## Introduction

The iterative bleaching extends multiplexity (IBEX) imaging method is an iterative immunolabeling and chemical bleaching method enabling highly multiplexed imaging of diverse tissues (*Radtke et al., 2020*; *Radtke et al., 2022*). IBEX is one of several multiplexed antibody-based imaging approaches offering insight into the cellular composition and spatial patterns of normal and diseased tissues (*Hickey et al., 2022*). While these methods are promising, there are several challenges associated with their adoption. These include the high cost of equipment and consumables and broad range of expertise required for sample preparation, antibody selection and validation, panel design, image acquisition, and data analysis (*Hickey et al., 2022*; *Quardokus et al., 2023*).

An additional challenge for adopting IBEX, and other multiplexed imaging methods, is due to limitations in the scientific publishing process. First and foremost, scientific publications offer static snapshots of a method and do not evolve with knowledge obtained from novel applications, adoption to new research settings, or improvements by others in the field. Even in instances where a detailed protocol exists (*Radtke et al., 2022*), adoption is not straightforward due to differences in the samples, reagents, hardware, and software that vary from those described in the original work. Furthermore, most publications do not include information about methodological decisions and reagents that did not work. As is common with scientific publications, investigators chiefly describe the path to success and omit failures, allowing others to blindly follow unsuccessful research paths. Finally, not all consumables and hardware components are available in all countries, and they may be discontinued, further complicating the path to adoption.

The need for sharing negative results is widely acknowledged across science (*Weintraub, 2016*; *Bespalov et al., 2019*; *Nimpf and Keays, 2020*; *Echevarría et al., 2021*). Unfortunately, it is still not practiced widely. Several attempts at providing dedicated publication venues for negative results have failed, either completely ceasing to exist or publishing less than ten manuscripts per year (*Nimpf and Keays, 2020*). Some attribute this behavior to a perceived low return on investment (*Echevarría et al., 2021*). The level of effort in terms of time invested in publishing negative results is likely to yield minimal returns in terms of citations. The question remains, what incentives motivate the sharing of negative data? An obvious incentive is direct financial rewards (*Nature Editorial, 2017*). This is the approach taken by the recently announced replication prize, September 2025, sponsored by the US National Institutes of Health. A slightly less direct approach is to lower the bar for data sharing, both in terms of time investment and minimal size of data contribution.

To address these challenges, we created the IBEX Knowledge-Base (KB), a central hub of knowledge for immunofluorescence-based methods. Originally focused on the IBEX method (*Radtke et al., 2024a*), we envision it expanding to methods such as CycIF (*Lin et al., 2015*), MIBI (*Angelo et al., 2014*), and others. Community is at the heart of the IBEX KB with a major focus on sharing solutions and advice at scale. Members are rewarded for contributing new knowledge, reproducing or refuting reagent validations, and reporting negative results. While the KB saves time and money for all researchers by preventing the pursuit of ineffective reagents, its impact is particularly significant for scientists in resource-limited regions, where access to reagents is constrained; the KB not only helps them identify viable options but also mitigates the cost of failed experiments, thereby maximizing their limited resources. Collectively, the KB and community built around it are reducing financial costs while increasing the pace of scientific discovery.

## Results
### Overview of supported methods, reagents, and metadata standards

The KB supports a wide range of multiplexed imaging methods. These methods include, but are not limited to, manual and automated IBEX methods (*Radtke et al., 2020*, *Radtke et al., 2022*), standard, single cycle multiplexed 2D imaging, amplification of low abundance targets with Opal dyes (Opal-plex) (*Radtke et al., 2020*), and use of the IBEX dye inactivation protocol with the Cell DIVE

**Table 1.** IBEX imaging KB directory structure and contents.

Infrastructure-related files are found in the data, docs_in, and root directories. Content of interest for the general user is found in the data and docs directories. Using text-based files and a data lake approach enables easy navigation, viewing, and editing of the raw KB content without requiring dedicated software.

| Directory | Content |
|---|---|
| data | • Files describing the contents of the knowledge-base, data dictionaries (reagent_data_dict.csv, reagent_glossary.csv). <br> • Files referencing external information (protocols.csv, videos.csv, vendor_urls.csv, publications.bib). <br> • Files referencing internal information (image_resources.csv, reagent_resources.csv). |
| docs_in | Markdown template files. |
| docs | Markdown files with notes and images that support claims made with respect to reagents. |
| .github | Configuration files for GitHub actions (automated testing and markdown file generation using the contents of the *data* and *docs_in* directories). |
| root | License, basic testing configuration via pre-commit and Zenodo configuration file listing contributor details (affiliation etc.). |

imaging platform (Cell DIVE-IBEX) (*Gerdes et al., 2013*; *Radtke et al., 2024b*). Beyond multiplexed methods applied to thin 2D (5–30 µm) tissue sections, we also support volumetric imaging of thick tissue sections (>200 µm to 3 mm) using the clearing enhanced 3D (Ce3D; *Li et al., 2017*; *Li et al., 2019*) and highly multiplexed 3D imaging methods (Ce3D-IBEX) (*Germain et al., 2022*). The community additionally supports a wide range of tissue preservation methods, for example, fresh frozen, fixed frozen, formalin-fixed paraffin-embedded (FFPE). Definitions of all supported imaging and tissue preservation methods are provided as part of the KB contents in the *reagent_glossary.csv* file (GitHub repository data directory, *Table 1*).

We have adopted the antibody metadata standards established by the Human Biomolecular Atlas Program (HuBMAP) (*Snyder et al., 2019*; *Jain et al., 2023*) originally reported by *Hickey et al., 2022* and later refined by *Quardokus et al., 2023*. These metadata fields include gene and protein identifiers established by the HUGO Gene Nomenclature Committee (HGNC; *Povey et al., 2001*) and the UniProt Consortium (*Bairoch et al., 2005*). A complete description of each field is found in the reagent_data_dict.csv (GitHub repository data directory, *Table 1*). We strongly encourage the reporting of Universal Protein Resource (UniProt) IDs (*The UniProt Consortium, 2008*) for each protein target and research resource identifiers (RRID) (*Bandrowski et al., 2016*) for reagents and tools whenever possible. This practice ensures accurate reporting of a protein target and its respective antibody. The KB allows scientists to find reagents for a wide range of target species utilized in basic and translational research, e.g., human, mouse, non-human primate, canine, zebrafish, etc. Members of the community report details known to influence antibody performance such as antigen retrieval conditions, the type and concentration of detergents in blocking and labeling buffers, target tissue, and tissue state, such as normal or malignant.

Multiplexed imaging experiments often require custom reagents created by the user or supplied by a company. For this reason, we distinguish between 'stock' reagents that can be readily incorporated into others' experiments versus 'custom' reagents generated by or for the user. An additional benefit to the community is the reporting of preferred conjugation kits for custom antibody solutions as well as a list of vendors that support custom solutions. Importantly, a flexible data structure allows the KB to evolve with the inclusion of diverse samples, reagents, and emerging techniques validated for multiplexed imaging. Finally, the reagent_resources.csv file lists information about the dye inactivation conditions for each fluorophore tested for cyclic imaging in 2D and 3D tissue volumes. These data are collated into the fluorescent_probes.csv file along with the spectral properties of each fluorescent probe evaluated. Both files are found in the *data directory* as described in *Table 1*.

## Knowledge-Base design: a three-faceted approach

The KB is organized as a data lake (*Fang, 2015*) where data are stored in original formats, jpg image files, comma-separated value (csv) files, JavaScript Object Notation (JSON) files, markdown template

files, and standard markdown files. The usage of simple text-based formats ensures that the raw files are both human and machine-readable, can be edited using many programs, and will be readable into the future. This guarantees that the data are interoperable, one of the key principles of FAIR data sharing (*Wilkinson et al., 2016*). Additionally, whenever possible, external information is referenced using unique and persistent identifiers. Antibodies are referenced using their RRIDs (*Bandrowski et al., 2016*; *Research resource identification portal, 2024*), contributors are referenced using their Open Researcher and Contributor ID (ORCID), and both publications and datasets are referenced by their Digital Object Identifiers (DOI). The raw information is stored in a directory structure that is easy to navigate and does not require any software installation or configuration to work with locally. The specific structure and contents are described in *Table 1*.

A key design decision made early on was that the KB will minimize the amount of internally hosted content by utilizing community-endorsed resources that are expected to exist long into the future, for example the Image Data Resource, Zenodo, YouTube. Content is stored internally only if no appropriate hosting services exist and is periodically archived in an external persistent data repository. All other contributions are first shared on existing hosting services and then reported on and aggregated via the KB. This minimizes the resource requirements for maintaining the KB and strategically places the KB into the existing open science ecosystem. Lastly, the contents of the KB are licensed under the

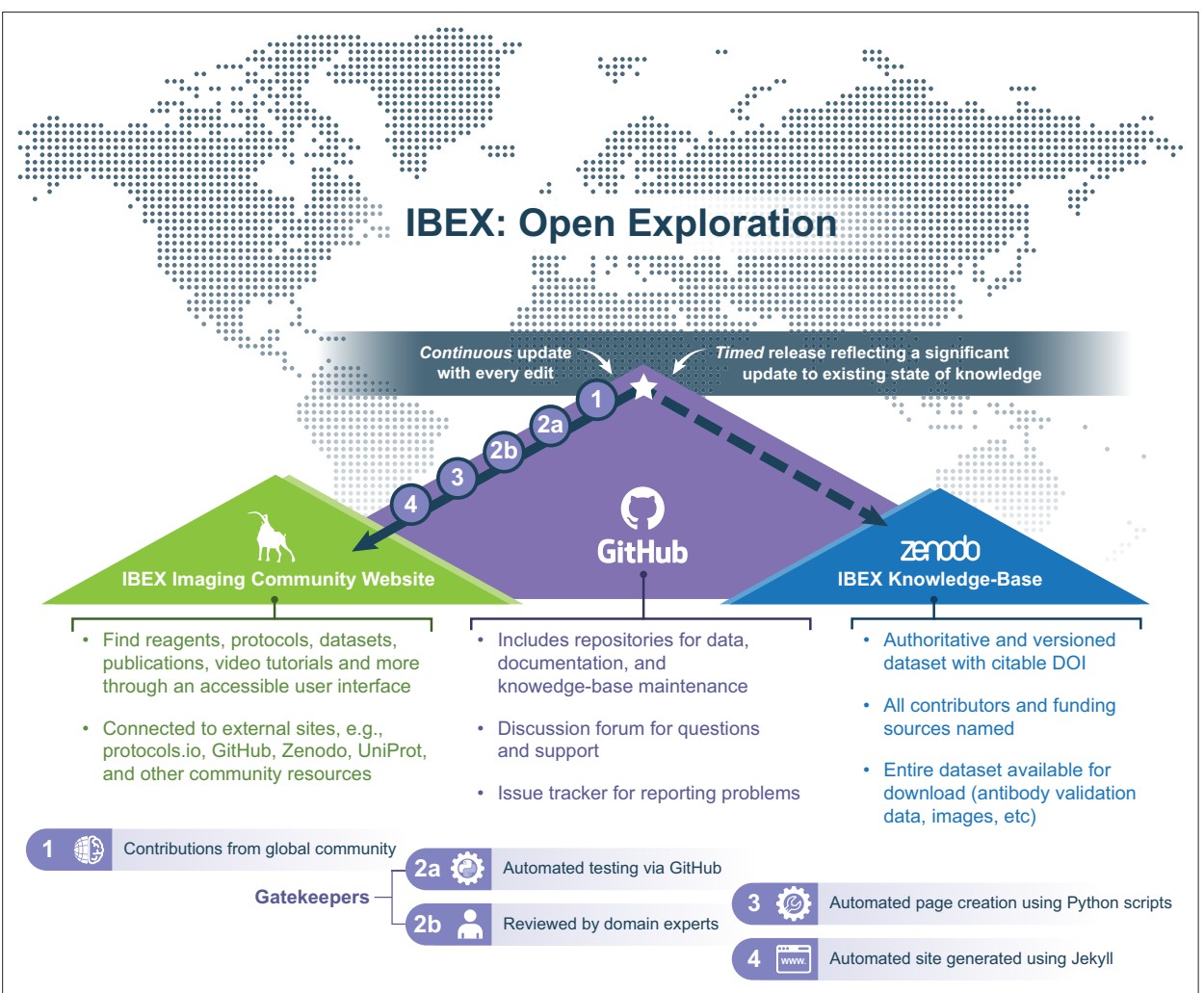

**Figure 1.** Schematic overview of IBEX KB design and contents. The IBEX KB enables open exploration of reagents, protocols, datasets, and advice related to multiplexed imaging from members of the global scientific community. The overall design consists of three facets: a GitHub source repository (purple mountain), a static website (green mountain), and a Zenodo data repository hosting citable archival versions of the data (blue mountain). Every addition to the KB initiates an update to the IBEX Imaging Community Website outlined in steps 1–4. In contrast, updates to the authoritative version on Zenodo are initiated by community members when a significant update to the state of knowledge justifies a new release.

**Figure 2.** Computational workflow for static website generation. Data displayed on the static website is generated from human and machine-readable comma-separated value (csv), bibliography database (bib), jpg images, and markdown template (M down arrow) files. Following a new submission, custom Python scripts expand the template markdown files to include the new data. Existing data (unprocessed markdown files) remain unchanged. The website is automatically generated via Python scripts utilizing the GitHub continuous integration compute infrastructure and static website creation services (Jekyll).

Creative Commons Attribution 4.0 International License. This permissive license allows others to freely use the data for both research and commercial purposes and only requires attribution.

Based on previous experience developing open-source software tools and establishing communities around them (*Enquobahrie et al., 2007*; *Yaniv et al., 2018*), we followed a similar approach here. As a result, the KB has three facets: (1) An online development platform providing a hosting service for repositories of source code and data as well as facilities for community interaction, (2) a static website, and (3) a service for hosting archival versions of the source/data. The most current version of the KB is available from a publicly accessible GitHub repository (https://github.com/IBEXImagingCommunity/ibex_imaging_knowledge_base; *Yaniv, 2026*). A user-friendly view of the current data is provided via an automatically generated static website (here). Finally, authoritative, citable, archival versions of the KB are created periodically and automatically shared on the Zenodo generalist repository (*Yaniv et al., 2023*). *Figure 1* provides an overview of these facets, each of which serves a distinct purpose as described below.

The use of a GitHub-hosted repository allows others to easily adopt and extend the KB for their own needs. A duplicate of the GitHub repository can be created at the press of a single button. Additionally, the GitHub ecosystem provides compute resources (used for automated data validation and website creation), tools for manual data review, issue reporting and tracking, website hosting, and a discussion forum. We have adopted the open-source software development stance: conducting discussions in the open and publicly resolving all issues, for example public history of issues and how they were addressed (see here). Once the KB GitHub repository is updated, the website automatically reflects these changes in a matter of minutes following the information flow shown in *Figure 2*. This website provides a user-friendly interface for browsing the current KB content.

The use of periodically archived versions of the KB on the Zenodo generalist data repository complies with FAIR data stewardship principles (*Wilkinson et al., 2016*). Furthermore, metadata associated with the Zenodo entry ensures the dataset is readily findable and accessible both for humans and computers via the Zenodo website and its associated application programming interface. The interoperability of the dataset is achieved by using simple and common file formats as described above. Additionally, the reuse and reproducibility FAIR principles are supported by the versioning mechanism of Zenodo, enabling one to refer to a specific version of the dataset to enable others to reproduce findings, as opposed to referring to the first or last version of the dataset when there is no versioning mechanism. Finally, usage of the Zenodo archiving mechanism enables us to officially provide credit to researchers who contributed to the KB. All contributors are acknowledged on the dataset's author byline, and there is a DOI associated with each version of the dataset. Unlike a publication that has a fixed list of authors, we envision a continually expanding author list. Consequently, there is a need to update the author byline and citation information, which is achieved by periodic archiving with a distinct DOI associated with each KB version.

## Contributing to the Knowledge-Base

Members of the scientific community can interact with the KB in one of two roles: a knowledge producer (contributor) or a knowledge consumer (user). *Figure 3* provides an overview of the various

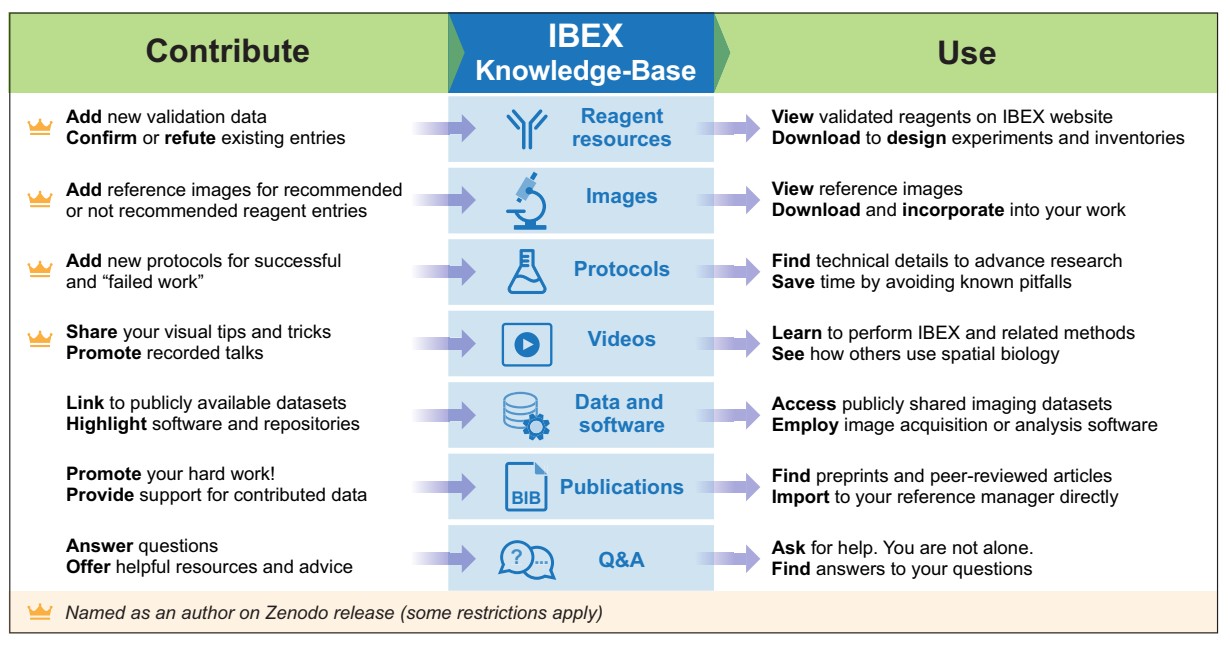

**Figure 3.** The IBEX KB provides several ways to contribute and use data. Summary of supported data types and ways to contribute or use the KB. Crown icon indicates contributions that result in authorship on the Zenodo archival versions.

interactions one can have in each of these roles. Guidelines for contributing information are summarized in *Figure 4* and detailed in the materials and methods section.

As a knowledge contributor, one can add information about external resources such as publications, datasets, software, protocols, and videos. If this content was created specifically for inclusion into the KB, we offer the contributor a place in the author byline. A different form of contribution that is critical for a thriving community is answering questions on the discussion forum. This facilitates sharing of solutions to commonly encountered problems and enables faster scientific progress. Finally, one can contribute to the internal resources associated with reagent validations, the key components of the KB.

A considerable amount of data is generated during the reagent validation and antibody panel design stages. These are time-consuming processes that also require significant financial investment (*Hickey et al., 2022*; *Quardokus et al., 2023*). Most publications and data repositories only include successful reagents with negative results only known to the persons directly involved in the work. For this reason, the KB includes results for validated reagents, both recommended and not, that is negative results. Contributors reporting positive, negative, and reproduced results are all rewarded with a place in the dataset's author byline. Thus, contributing new data requires much less effort than a journal publication while still providing benefit to the contributor in terms of authorship. We require supporting material for all contributed reagents as summarized in *Figure 4* and detailed in the materials and methods section.

Several practices for antibody validation have been previously described (*Bordeaux et al., 2010*; *Uhlen et al., 2016*; *Hickey et al., 2022*). These include evaluating the immunolabeling pattern of a particular antibody with positive and negative controls, assessing colocalization with orthogonal markers, and using knockout or knockdown cell lines. We appreciate the time it takes to submit a KB reagent validation entry. For this reason, we elected for a validation standard that encourages rigor without being oppressively burdensome. Furthermore, the KB is designed to be self-correcting, allowing entries to be refined as more people contribute their data. An ideal entry includes validation images showing the labeling pattern of an antibody in positive and negative control tissues. Alternatively, the image(s) may depict the proper subcellular and tissue distribution of the antibody. Additional details on the controls used, colocalization with other markers, and descriptions of the cellular and subcellular distribution of the antibody (membrane, cytoplasm, nuclei) can be included in the notes section of the supporting material file. If images were not captured at the time of testing, we accept

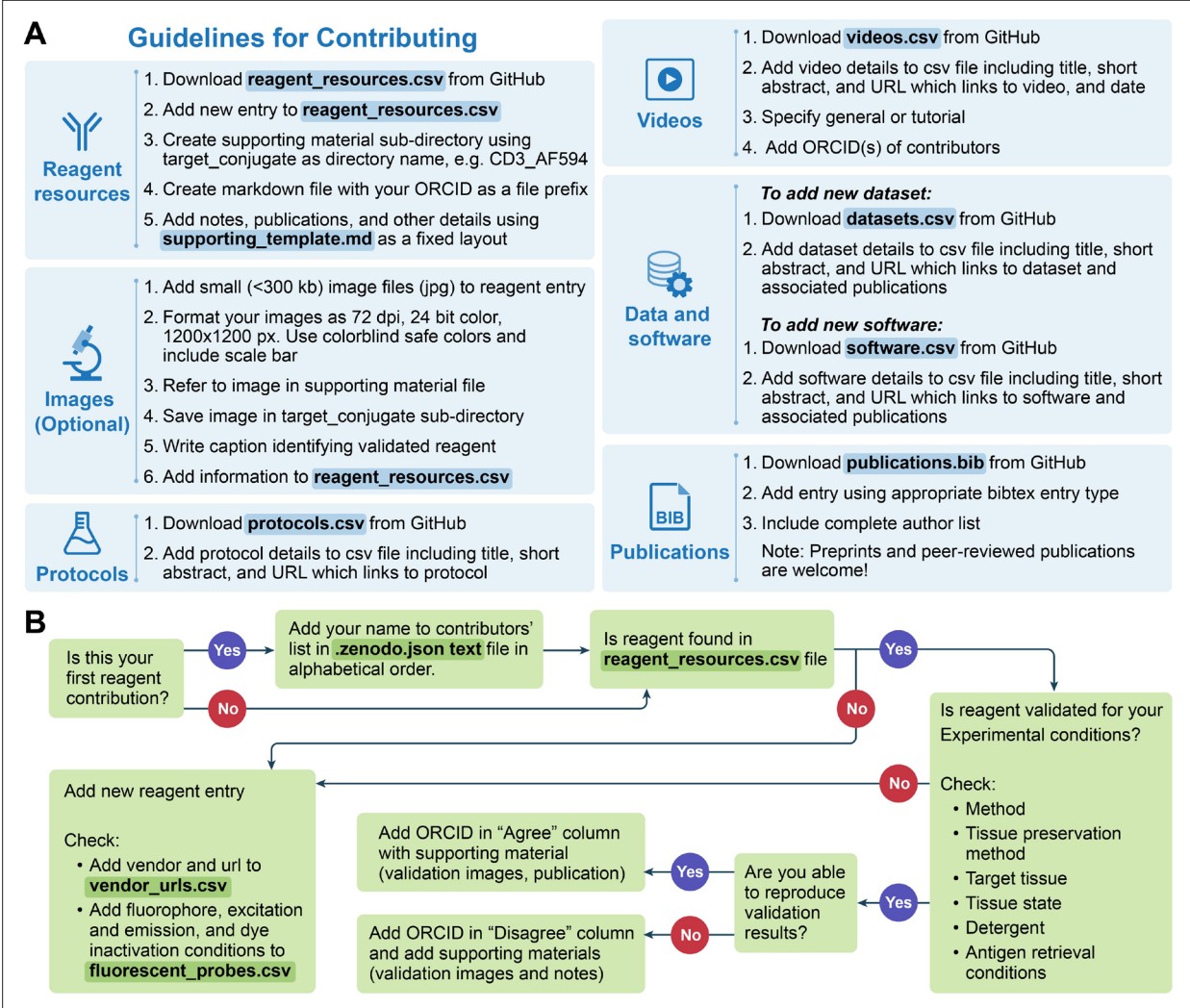

**Figure 4.** Guidelines for contributing data and flow chart detailing how to add a reagent contribution. (**A**) Details for contributing data to the IBEX KB. Files that need to be modified are highlighted in blue with data file name bolded. (**B**) Flow chart demonstrating how to add a reagent contribution. Files that need to be modified are highlighted in green with data file name bolded.

entries that describe why an antibody is recommended or not recommended for a particular target. Oftentimes, an antibody works better in a particular conjugate or tissue. We encourage contributors to provide such details. For reagents published in peer-reviewed manuscripts, the supporting materials can link to an appropriate publication.

The 'game' of science is often viewed as self-correcting. This does not necessarily happen without an explicit effort (*Ioannidis, 2012*). To facilitate timely self-correction, we encourage reproduction of reagent validations, both of positive and negative results. For each validation configuration, we have three key entries: (1) the original contributor's identity; (2) a list of up to five individuals that reproduced the validation and *agree* with the recommendation; and (3) a list of up to five individuals that reproduced the validation and *disagree* with the recommendation. Initially, the original contributor is also listed as the first individual to agree with the recommendation. All individuals are uniquely identified by their ORCIDs. In this manner, we can identify both positive and negative reagent configurations that were confirmed by multiple independent scientists, preferably from different laboratories, providing us with confidence in the results. In some cases, there are disagreements between scientists. When the number of scientists disagreeing with the original recommendation is greater than those agreeing with it, the failed replication study contribution is opened for public discussion on the KB forum before acceptance. This allows the original contributor and other members of the community to engage with the researchers who were unable to replicate the specific validation. We expect

such discussions to either expose missing details that are required for replication, adding them to the supporting material, or to identify and document issues with the original work. In the latter case, the recommendation is overturned and the ORCIDs in the agree and disagree categories are swapped (A first disagreement was recently highlighted on the KB discussion forum.). An overview and detailed reagent validation contribution workflow are given in *Figure 4*.

Contributing information into the KB is done using the git version control system and the GitHub repository hosting service. This requires a basic understanding of both and following the detailed instructions for contributing. Community members who cannot install the relevant software or feel uncomfortable using it can download an archive of the KB, modify it locally, and then reach out to the KB maintainers for help sharing their contribution. The submission is then completed together, with the maintainers dealing with the usage of git and GitHub.

## Using the Knowledge-Base: several use cases

There are several ways to use the KB. The first and most common usage is to search for a reagent using the 'Reagent Resources' tab on the IBEX Imaging Community static website and searchable drop-down menus associated with each metadata field. Using the filter function, one can identify antibodies recommended by the community, identify the optimal working conditions for the indicated reagent, and gain insight into the cell types and anatomical structures labeled by a particular reagent for the specified tissue and tissue state, for example normal, malignant, or infected with *Mycobacterium avium* or Ebola virus (EBOV). For example, an antibody against CD11b (https://ibeximagingcommunity.github.io/ibex_imaging_knowledge_base/supporting_material/CD11b_AF647/0000-0001-9561-4256.html) is recommended for human tonsil FFPE tissue sections following antigen retrieval with a pH 9 buffer. In contrast, another member of the community reports that the unconjugated version of this antibody does not work in human tonsil FFPE tissue sections following antigen retrieval with a pH 6 buffer (https://ibeximagingcommunity.github.io/ibex_imaging_knowledge_base/supporting_material/CD11b_Unconjugated/0000-0003-4379-8967.html). These results underscore the impact of the antigen retrieval buffer on antibody labeling as discussed in the forum. For greater exploration of the data, the *reagent_resources.csv* file can be downloaded from GitHub and analyzed with a variety of open source and commercial software.

The KB includes images supporting many of the reagent validations; these can serve as a basic quality assurance tool, a reference for the expected quality of results. The quality of new images acquired by researchers, other than the original data contributor, is expected to be as good or better than these reference images for the specific reagents and tissue types. If there is a significant discrepancy, we expect researchers to inquire about it on the KB discussion forum, asking the community for help in identifying the possible reasons for such differences.

As the KB includes information about protocols, data, and software, it can help others find alternatives to discontinued, expensive, or unavailable consumables. For example, *Moriarty et al., 2024* empowered the community by creating and sharing a solution for a discontinued piece of hardware on the NIH 3D repository. In addition, *Fritzsche, 2024* optimized a protocol for chrome alum gelatin, the preferred adhesive for the IBEX Imaging Community. In the original IBEX manuscript (*Radtke et al., 2020*, *Radtke et al., 2022*), this adhesive was purchased from a vendor that does not ship outside the U.S. Importantly, these acts of goodwill are associated with digital object identifiers minted by the data repositories NIH 3D and Protocols.io, enabling others to cite their contributions as done here.

Contributing information into the KB can also facilitate streamlining of the publication process by providing validation data, detailed protocols, video tutorials, and links to associated datasets and software. To date, the KB has been cited in the methods of several manuscripts (*Radtke et al., 2024b*; *Yayon et al., 2024*) and was used to address reviewer comments related to antibody validation and multiplexed tissue imaging (*Radtke et al., 2024b*). While not as altruistic as the other use cases, it is a welcome reward for the upfront investment of contributing.

Finally, the KB can be used to provide up-to-date, domain-specific, context for AI agents with the goal of answering questions and performing tasks relevant to multiplexed imaging, such as analyzing the contents of the KB or designing imaging panels. This form of interaction can be done by either downloading a copy of the KB files to a local computer and providing the files directly to the AI agent, or by pointing the AI agent to the KB website. The interaction between an AI agent and external resources such as the KB website is made possible via the open Model Context Protocol (*Anthropic,*

*2024*; *Hou et al., 2025*) which enables the AI agent to fetch and analyze the contents of the site. The resulting retrieval-augmented generation of text (*Lewis et al., 2020*; *Brown et al., 2025*) utilizes this up-to-date domain-specific information and is likely to provide more relevant responses with fewer errors (reducing model hallucinations). Note that AI responses require subject matter supervision as they may contain incorrect statements. See supplemental material for an example conversation with an AI agent using the KB website as context.

## Citing the Knowledge-Base

Most often, resources are cited by referring to the manuscript that introduced them. This is a valid approach when the resource is static and all contributors are listed on the original manuscript. In our case, we are describing a dynamic resource with unknown future contributors. Therefore, when referring to the general concept of the IBEX KB, we request that the relevant publications and the KB's Zenodo concept level DOI, that is *Yaniv et al., 2023*, be cited. In all other cases, we request that the relevant publications be cited and the specific version level DOI (e.g. *Yaniv et al., 2024* for version v0.2.0). This citation approach satisfies two guiding principles of the IBEX imaging community, commitment to excellence and shared ownership. By citing the version level DOI, you enable others to reproduce the work based on the same information available when you used it, and you give credit to all relevant contributors, including those who contributed knowledge after the original manuscripts describing the KB were published.

## Discussion

The IBEX KB is an open, global repository that provides information related to IBEX and other 2D and 3D immunofluorescence imaging methods. Unlike manuscripts that only provide a static snapshot of a method, the KB empowers method adoption and evolution by providing a dynamic resource for reagents, protocols, datasets, software, and more. To achieve this goal, the IBEX KB facilitates the practice of open science by enabling the public sharing of all outputs of the scientific endeavor. By working together as a community, we aim to advance scientific discovery at a faster pace and in a more efficient manner. An additional benefit of open science is increased research integrity (*Haven et al., 2022*) and greater public trust in the scientific process (*Rosman et al., 2022*), a non-trivial challenge (*Agley, 2020*).

The shift towards open science has been an ongoing process over the past two decades. Initially, it started as a grassroots effort reliant only on internal motivation; however, in recent years, open science practices have become a requirement via mandates from funding agencies with respect to sharing various outputs created as part of the scientific process (*National Academies of Sciences, Engineering, and Medicine, 2018*; *Bertram et al., 2023*; *Cobey et al., 2023*). While mandates are important, we see internal motivation as the key to practicing open science. Most importantly, we note that by practicing open science, you are not only helping others, you are helping 'future you' retain all of the detailed knowledge 'current you' possesses. It is not uncommon to document information for ourselves in a less detailed manner than we would if we intended to share it with others. Particularly, this pertains to details that are currently obvious in one's mind and hence not explicitly documented. As time goes by, natural memory decay occurs, and what was once obvious is no longer so (*Davis and Zhong, 2017*). While following the open science philosophy is desirable, previous studies have identified barriers towards its adoption (*National Academies of Sciences, Engineering, and Medicine, 2018*; *Zuiderwijk et al., 2020*; *Gomes et al., 2022*).

Following open science practices throughout the immunofluorescence imaging research life cycle, we review associated barriers and highlight how our community addresses them with an emphasis on providing internal motivation to adopting this research philosophy. First, before performing IBEX, one must identify and validate appropriate reagents for specific experimental conditions. This is the one element in the IBEX research life cycle that had no existing community-endorsed hosting service and is thus internal to the KB. The time, cost, and expertise required for validating antibodies and other reagents for multiplexed imaging is well documented (*Hickey et al., 2022*; *Quardokus et al., 2023*). Furthermore, it is widely acknowledged that sharing both positive and negative antibody validation has the potential to greatly reduce research costs for other researchers. Despite these benefits, some individuals may be hesitant to contribute as it provides an advantage to potential

competitors, particularly negative results that are traditionally not reported anywhere. While this is true, we believe that openly sharing this information is better for the scientific community as a whole, and we encourage the practice by adding contributors of reagent validation results, positive or negative, to the author byline.

Once the validated reagents are publicly shared, we turn to detailed experimental steps for reproducible experimental workflows, the protocols. We encourage publishing detailed protocols both in dedicated peer-reviewed journals and non-peer-reviewed venues such as protocols.io. In both cases, the venue facilitates citation of the protocol and enables corrections via the journal error correction mechanism or the protocols.io versioning mechanism. While writing a detailed protocol requires a significant investment of time for the contributors, it serves to document the experimental workflow in sufficient detail so that others can replicate it. While a written protocol is useful, there are details that may be unintentionally omitted or are not described in a sufficiently clear manner or at all. In such cases, we recommend providing short video tutorials to highlight steps that are hard to describe using natural language. These videos can be shared using existing hosting venues such as the KB YouTube channel. Useful videos do not require a significant time investment and can currently be acquired and edited using a cell phone and free editing software such as Clipchamp (available on all Windows 11 systems).

Having shared the experimental protocol, we turn our attention to sharing its output, the acquired image datasets. In terms of data hosting, there are many free data hosting services, both domain-specific and generalist repositories. For the IBEX imaging community, members have deposited data both in domain-specific repositories, the Image Data Resource (*Williams et al., 2017*) and The BioImage Archive (*Hartley et al., 2022*), and in the Zenodo generalist repository (*European Organization For Nuclear Research, 2013*). While sharing the data requires additional curation efforts, there are benefits to investing the time collating all of the metadata information associated with the images in an organized fashion. This effort increases one's confidence in the data itself, as we invest more time in scrutinizing it, which in turn improves research integrity (*Haven et al., 2022*). Additionally, sharing the data shows that we are sufficiently confident in it. A practice that should assure more junior researchers that not all data are pristine, void of all artifacts, and that this data has utility. We recommend depositing data in repositories that provide DOIs over those that do not. This enables direct data citation, allowing the community to separate between the utility of a dataset from the utility of the scientific work in which it was originally acquired. We also prefer data repositories that support data versioning. This enables correction of errors, if and when they happen, and improves the ability to reproduce studies that rely on the shared data. Finally, as we are in the era of artificial intelligence (AI) (*Liu et al., 2021*), publicly available data has the potential to contribute to the development and evaluation of AI algorithms as we gain a broader view of image variations across experimental settings. Case in point, 38% of the data utilized in training the spatial proteomics foundation AI model described in *Shaban et al., 2025* are publicly available IBEX images from previous studies. It should be noted that there was no connection between the data producers and the groups that developed the AI model, illustrating how publicly sharing data enables scientific progress.

We next focus on depositing the software used to create, analyze, or process the imaging data. The level of effort required for contributing new software is expected to be minimal. The expectation is that the software be reasonably documented for others to use. This does not imply good design or optimal implementation. These are not requirements from general research software. The main principle is that sharing the software promotes transparency and thus trust in the scientific process. Ideally, the software is provided as open-source with permissive usage licenses, but this is not a requirement. Open-source software sharing is expected to use common hosting services such as GitHub, GitLab, Bitbucket, or other publicly accessible services. A key requirement, which is often not implemented by scientists who write short analysis scripts, is to assign them a version. This is trivial to do using the hosting services to create code releases and should be done even if the researcher only expects to share a single release. Once the software is shared via a public hosting service, the KB should be updated to reference it.

With the software shared, all that remains is to share the culminating artifact of the research life cycle, the resulting publication. Many journals are either fully open access or offer an open access option with the associated higher costs shouldered by the authors as compared to the closed access option. Our community prefers publishing in an open access fashion when the funds are available, as

this has been shown to increase citations (*McKiernan et al., 2016*), but closed access is also acceptable. The expectation is that no matter the access type, authors share their work as early as possible on the relevant pre-print server, i.e. arXiv, bioRxiv, or medRxiv. This has no financial costs and ensures timely dissemination of knowledge, sometimes years before the peer-reviewed manuscript appears in press. Additionally, all these pre-print servers support versioning, allowing the authors to update the publication as needed. Once the manuscript is published, peer-reviewed or not, the KB is updated with a reference to it.

An additional component of the KB is the dedicated discussion forum. We believe this functionality helps us improve the KB by highlighting missing information and by sharing information in an archival manner. That is, once a question is answered, there is no need to repeat the answer if someone else is facing the same challenge. The benefits of a dedicated discussion forum are not just transactional, a person asking a question and receiving an answer. For the person asking a question, it is not uncommon that they feel they are working in isolation and no one in their local research group can help them, which is why they reach out to the community. By asking on the public discussion forum and receiving an answer, there is the immediate benefit of obtaining a solution to the particular problem. Interestingly, there is also a potential for additional psychological benefit, feeling that you are working with others on the same problem. These types of social cues have been shown to increase intrinsic motivation even when people are working alone (*Carr and Walton, 2014*). Similar to the person asking the question, the person answering the question may obtain psychological benefits from this altruistic act, as it has been shown that providing help to others is beneficial for one's own health (*Poulin et al., 2013*; *Hermanstyne et al., 2022*).

To sustain the KB growth efforts, its chairs meet bi-weekly to ensure continued development and maintenance. In addition to these regular meetings, we engage with both current and prospective community members to gather feedback, encourage contributions, and expand the collective knowledge supporting the KB. To broaden outreach and foster sustained engagement, the IBEX community will collaborate with synergistic initiatives such as the HuBMAP Affinity Reagents Working Group, the European Society for Spatial Biology (ESSB), and the Global Alliance for Spatial Technologies (GESTALT).

As a further incentive for participation, we intend to launch an annual 'Reagent Validation Week', a community-driven event inspired by software hackathons. During this dedicated week, researchers would focus on validating or reproducing validation for selected reagents and contribute their findings to the KB. We have also discussed hosting an 'Around the World' symposium, featuring presentations from both junior and senior scientists across the community, to showcase diverse perspectives and foster global collaboration.

In summary, the IBEX KB empowers the practice of open science throughout the research life cycle by providing community-validated reagents, protocols, datasets, and software related to 2D and 3D immunofluorescence imaging. While the initial focus was on IBEX, the current state of knowledge has evolved to include other imaging methods beyond IBEX such as Ce3D, Ce3D-IBEX, and Cell DIVE-IBEX. Importantly, the KB is designed to evolve with the current state of knowledge. We are optimistic that future versions will include extension of the IBEX method to other tissues and species, and we intend to solicit contributions of reagent validations for other multiplexed imaging techniques such as CycIF (*Lin et al., 2015*). At that point in time, we expect to re-brand the KB as the *IBEX ++Knowledge* Base, indicating that it is the extended version of the original KB. The name maintains ties to the original technique which sparked the KB creation and our usage of practices from the software development world, echoing the C++origin story (*Stroustrup, 1996*).

## Materials and methods

Individuals can directly interact with the KB in one of two roles: a knowledge producer, contributor, or a knowledge consumer, user. *Figure 3* provides an overview of the various interactions one can have in each of these roles.

### Prerequisites for starting: ORCID and optionally a GitHub account

Contributions to the KB can follow one of two paths. The primary path involves usage of the git version control system and the GitHub hosting service. This path has two prerequisites, obtaining an

ORCID and a GitHub account. The secondary path is similar in terms of data preparation but does not require that the contributor use git and GitHub. Thus, the only prerequisite is an ORCID.

Following the primary path, there is a one-time setup step for first-time contributors. If comfortable working from the command line, install the git version control software on your computer. Otherwise, install the GitHub Desktop software that includes git and provides a convenient graphical user interface. First, go to the KB GitHub repository and create a copy of the KB under your personal account. This is referred to as forking and is done by clicking the button that says 'fork'. After completing this step, clone the personal GitHub copy of the KB to your local machine. This creates a local copy of all the files that can then be safely modified. Finally, add the contributor information (name, affiliation, and ORCID) to the 'creators' section in the hidden *.zenodo.json* text file. On a Windows system, open File Explorer and select View-Show-Hidden items. Additions to this file are required to maintain alphabetical order based on last name and be inserted between the first and last two entries; these three entries remain in their fixed locations in the list. By adding the details to this file, the contributor will be automatically listed in the Zenodo author byline when the next release that includes their contributions is made.

Following the secondary path, all git and GitHub-related steps listed in the contributing instructions are ignored by the contributor. Instead, the contributor downloads a zip archive file containing the current version of the KB, edits the contents as described below, and shares the updated files with the KB maintainers who complete the contribution using git and GitHub. As this path does not require the contributor to use GitHub, communication between the contributor and KB maintainers is done via email and is not visible to the public. To maintain transparency, as part of the contribution review process, the KB maintainers will post relevant responses provided by the contributor on GitHub (akin to a journal's open review process). The posted questions and responses from the first contribution following this path are available on GitHub (here). In this case, changes proposed by the subject matter expert reviewing the contribution were reverted based on the contributor's response.

While many from the biological sciences may shy away from git and GitHub, preferring the secondary path to contributing, we highly recommend reading (*Braga et al., 2023* and *Chen et al., 2025*). These publications show that git and GitHub are useful tools for general laboratory research and that they have the potential to accelerate research while improving reproducibility. Learning to use these tools may be worth the effort even if one is far removed from the realm of software development.

## Adding protocols, videos, datasets, software, and publications

To add value to the scientific process while utilizing existing, well-established resources, the KB does not directly store protocols, videos, datasets, or software. To contribute these forms of scientific knowledge, contributors are expected to first deposit them in well-established venues and then update the KB reference information. The KB thus aggregates information from multiple external resources with minimal duplication. This information is stored in a set of CSV files and a bibliography file in the text-based BibTeX format.

The KB accepts references to external protocols including both peer-reviewed protocols, for example, protocols published in Nature Protocols, STAR Protocols, Nature Methods, JoVE, etc. and non-peer-reviewed protocols such as those shared via protocols.io. The former have the advantage of benefiting from the peer review process, resulting in clearer instructions. The latter have the potential advantage of including detailed descriptions of approaches that were tried and did not work, reporting negative results that are not usually included in peer-reviewed protocols. Once publicly available, add the protocol information to the KB. Edit the *protocols.csv* file and add the protocol title, URL, and optionally a short abstract describing the protocol in the *Details* column. If the abstract requires richer formatting than plain text, use HTML formatting. Do not use non-ASCII characters in the title or description. Represent them using standard ASCII characters, for example instead of $\alpha$ write alpha.

The KB accepts references to external videos hosted on freely accessible video hosting services, for example, YouTube, Vimeo, etc. If desired, the contributor can provide the video to the KB maintainers for upload to the IBEX Imaging Community YouTube channel. Once uploaded to the hosting service, the contributor adds the video information to the KB. Edit the *videos.csv* file and add the video title, URL, category (general or tutorial), date, and optionally a short abstract describing the video in the *Details* column and the ORCIDs of the contributors in the *Contributors* column. If the details require

richer formatting than plain text, use HTML formatting. When a video was created by multiple contributors, list the ORCIDs separated with a semicolon, for example *0000-0003-4379-8967;0000-0003-0315-7727*. Note that the researcher details for the listed ORCIDs must appear in the.zenodo.json file.

The KB accepts references to external datasets that are expected to be deposited in freely accessible data repositories. When appropriate domain-specific repositories exist, data should be deposited there, for example, The Image Data Resource (*Williams et al., 2017*), The BioImage Archive (*Hartley et al., 2022*). Otherwise, deposit the data in a generalist repository, for example Zenodo, Figshare. Selecting a repository requires considering several criteria. These include limits on file size and total dataset size, file format requirements, whether the repository provides a versioning system, the level of effort required to satisfy the repository's requirements on accompanying metadata details, and whether the repository provides a DOI for the dataset to enable others to cite it. Once deposited in a data repository, add the dataset information in the KB. Edit the *datasets.csv* file and add the dataset title, URL, and year. Optionally, add a short abstract describing the dataset in the *Details* column, and the dataset license and associated publication DOIs in the corresponding columns. If the details require richer formatting than plain text, use html formatting. If there is more than one associated publication, list them using a semicolon as a separator, for example *10.1038 /s41596-021-00644-9;10.48550/arXiv.2107.11364*. This is useful in cases where the preprint version of a paper is accessible without restrictions and the final journal paper is behind a paywall.

The KB accepts references to external software that is freely available from online sites such as GitHub, Zenodo, or other standard software hosting services. Ideally, software contributions to the KB are provided as open source and are accompanied by representative datasets and possibly videos illustrating usage in both straightforward and challenging cases. It should be noted that the companion datasets often differ from full-sized datasets required for research reproducibility and may be more appropriate for sharing using generalist data repositories. These datasets are provided to facilitate user understanding of correct software usage. That is, one utilizes these datasets as input to the software to obtain known results prior to attempting to apply the software to one's own data. To list software on the KB, edit the *software.csv* file and add the software title, URL, year, and license. Optionally, add a short abstract describing the software in the *Details* column and the software source code repository URL, programming language, and associated publication DOIs in the corresponding columns. If the details require richer formatting than plain text, use HTML formatting. If there is more than one associated publication, list them using a semicolon as a separator.

To add an entry to the bibliography file, *publications.bib* use the appropriate bibtex entry type and include the complete list of authors. As part of the KB requirements, the bibliography entry must include the publication's DOI and the note field that lists the corresponding authors.

## Adding or modifying a reagent resource

The KB supports any reagent used for multiplexed tissue imaging, for example primary antibodies, secondary antibodies, nuclear dyes, blocking kits, conjugation kits, lectins, and more. To add or modify reagent information, start by editing the *reagent_resources.csv* file followed by providing the required supporting material. *Figure 4B* shows the workflow for adding a new reagent resource or modifying an existing one. When modifying the file, do not use non-ASCII characters. Represent them using standard ASCII characters, for example instead of $\alpha$ write alpha. If using Excel or Google Docs to edit the file, you need to be careful if your preferences include automatic data conversion, as this may change the file content incorrectly. In our case, a conjugate such as 9E10 will be converted to a number, 90000000000. In Excel, disable the 'Automatic Data Conversion' before opening the file. In Google Docs, open an empty spreadsheet and then *import* the CSV file. During the import processes, first select the delimiter, comma in our case, and then specify the column type as 'text' for all columns.

Before adding a new reagent, check if it has been validated by members of the community by using the dropdown filters in your editor. If no reagent entry exists for your experimental conditions (species, tissue preservation method, antigen retrieval conditions, etc), add a line to the file corresponding to this new reagent entry. For antibodies directed against protein targets, please add species-specific UniProt IDs.

Whenever possible, include RRIDs with each reagent entry. RRIDs are present on many vendor pages or can be queried by catalog number and vendor on SciCrunch.org. If an RRID is not included, consider registering it via the Research Resource Identification Portal or list 'NA' if not available.

Additionally, check if the vendor and fluorescent conjugate of the new entry are included in the *vendors_urls.csv* and *fluorescent_probes.csv*. If they are missing, add the details to these files. If the contribution is reproducing a listed experimental condition, add your ORCID to the 'Agree' column if you were able to replicate the results. If you are unable to reproduce the findings, then add your ORCID to the 'Disagree' column. In both cases, use a semicolon as a separator, for example ORCID1; ORCID2; ORCID3. The KB allows for up to five ORCID values in each of these columns. This means that the original contributor's work was replicated by up to four people from other laboratories or refuted by up to five people from other laboratories. All reagent contributions, new or reproduction results, require additional supporting material.

## Reagent resources supporting material

The supporting material takes two forms: text files in markdown format and optional image files. All files are organized in a subdirectory structure using a target_conjugate schema as the directory name (see docs directory in GitHub for examples). When creating the target_conjugate subdirectory you will need to replace the following characters with an underscore: space, tab, /, ", {, }, [, ], (,), <, >,:, &. Consequently, supporting material for the target conjugate pair of 'Chicken IgY (H&L)', 'FITC' is found in a directory named 'Chicken_IgY__H_L__FITC'. Finally, the file name is based on the contributor's ORCID, for example 0009-0000-2047-4228.md. This text file consists of three sections, configurations, publications, and additional notes. The configurations section lists the experimental conditions, similar to the information found in the *reagent_resources.csv* file for the particular contributor, in addition to including internal links to the publications and additional notes sections for each configuration. The use of free-form text in the notes and publications sections allows sharing of detailed information about a particular reagent such as the optimal concentration for a particular tissue or method, known sensitivity to dye inactivation conditions, and recommended placement in an antibody panel. Here is an example of the kind of content that can be included in the notes section: 'Evaluated in human tonsil FFPE samples with HLA-DR (clone LN-3) antibody. Does not label myeloid or follicular dendritic cells in the lymph node. Recommend Abcam ab241408 (clone SP330) or Abcam ab238794 (clone SP331) in place of this antibody.' The author highlights details related to the tissue (human tonsil FFPE), approach (assessment of co-labeling with validated antibody), results (does not label anticipated cell types), and recommendations for other antibodies or conjugates. In summary, the markdown file allows the sharing of details commonly recorded in laboratory notebooks, but not easily reduced to machine and human-readable fields.

While optional, we strongly encourage the inclusion of images with reagent entries. To include an image with a reagent entry, first format images as 72 dpi, 24 bit color, and 1200×1200 px. Please include a scale bar, use color blind safe colors (Cyan, red; magenta, green, blue; no green and red), and capture at a sufficient zoom to allow fine details to be easily accessed. Next, refer to the image in the corresponding supporting material file and save it in the appropriate target_conjugate subdirectory, for example CD11b_AF488 for the example below. Next, write a caption using the following template: Species tissue marker (color, catalog number) as shown here: Mouse tumor: CD11c (cyan, catalog number 117312) and CD11b (yellow, catalog number 101217). The last step is to obtain the MD5 hash for the image file using the relevant program on your operating system (e.g. Windows Powershell CertUtil -hashfile myfile.jpg MD5). This unique value ensures data integrity, the contributed image was not modified during upload. The last step is to add information about the image to the *reagent_resources.csv* file under the columns titled 'Image Files', 'Captions', and 'MD5'. If there is more than one file associated with the row, separate the file names, captions, and MD5 hashes using a semicolon. On the static website, the images are directly accessible as part of the reagents table. Note that they are similar to images you would use in a manuscript and are not a substitute for publicly sharing the original microscopy images and metadata on domain-specific or generalist data archives.

Finally, when following the primary contribution path, share the information using git. First, create a new local git branch based off the 'main' branch and commit all changes into it. Then push the branch to your personal copy of the KB on GitHub. Lastly, on GitHub, create a pull request indicating to the KB maintainers that there is a potential new contribution for acceptance into the KB, automatically initiating data validation. When following the secondary contribution path, these steps will be carried out by the KB maintainers.

## Data validation

To ensure that contributed data complies with KB requirements and all relevant information is included, we implemented a two-step validation system that includes automated testing followed by manual review.

The first step consists of fully automated testing and utilizes the GitHub compute infrastructure to run data validation scripts ensuring that the syntax and content of the contributed files are valid. If automated testing fails, the system notifies the contributor and KB maintainers, and the contributor is expected to resolve the identified issues. At this point, the contributor can also interact with the KB maintainers and seek their help addressing the issues. Once automated testing passes, the second validation step starts. A subject matter expert reviews the submitted files and can interact with the contributor to address any reservations they have with respect to the contribution using the GitHub messaging system. Once the subject matter expert approves the contribution, it is merged into the KB.

Automated testing includes validation of data formatting and data consistency across the KB. This includes validating that the contributor information, vendors, and validation configurations listed in the reagent_resources.csv file are consistent with those listed in the .zenodo.json, vendor_urls.csv files, and the supporting material files. All CSV files are validated to ensure that columns that are required to contain information do indeed include it. For example, the datasets.csv file requires that there be content for the *Title*, *URL,* and *Year* columns, but this is optional for the *Details*, *Associated Publication DOIs,* and *License* columns. Data that is required to be unique within a file is also automatically checked. For example, the citation keys in the publications.bib, the ORCIDs in the .zenodo.json file, and the *URL* column contents in the protocols.csv file. Additionally, images that are shared as supporting materials are listed in the reagent_resources.csv file alongside their unique md5 hash that is used as a checksum to verify the integrity of the uploaded image. As many of the general validation concepts are similar across data files but differ in the specifics, the validation scripts utilize JSON configuration files to customize the specific validation tests per data file. All test configuration files are included in the KB git repository. The validation scripts source code is publicly available from a separate GitHub repository (https://github.com/IBEXImagingCommunity/ibex_imaging_knowledge_base_utilities; *Yaniv, 2025*) under the permissive Apache 2.0 license, allowing for free commercial and academic usage.

## Community and code of conduct

As a community, we have certain expectations with respect to the way we interact with each other. These have been formalized, with the community adopting the contributor covenant code of conduct version 2.1 as our guiding document. The reason for adopting a formal code of conduct is that the community is diverse. Members are at all stages of the academic career, from very junior to very senior researchers, and come from multiple countries across the globe. Thus, behavior that is acceptable in one place is not in another. Such cultural differences have the potential to result in uncomfortable situations. The formal code of conduct is expected to help minimize such situations.

## Acknowledgements

We are deeply appreciative of Arlene Radtke for her encouragement, proof-reading, and assistance with antibody metadata fields. This research was supported [in part] by the Intramural Research Program of the National Institutes of Health (NIH). The contributions of the NIH author(s) were made as part of their official duties as NIH federal employees, are in compliance with agency policy requirements, and are considered Works of the United States Government. However, the findings and conclusions presented in this paper are those of the author(s) and do not necessarily reflect the views of the NIH or the U.S. Department of Health and Human Services. ZY is supported by the BCBB Support Services Contract HHSN316201300006W/75N93022F00001 to Guidehouse Digital. This work was supported by the Division of Intramural Research, NIAID, NIH, Center for Cancer Research, NCI, NIH, NHLBI, NIH, and NIAMS, NIH. KB is supported by the NIH Common Fund through the Office of Strategic Coordination/Office of the NIH Director under award OT2OD033756, the SenNet CODCC under award number U24CA268108, NIDDK under award U24DK135157, the KPMP grant U2CDK114886, and the CIFAR MacMillan Multiscale Human program. CJC is supported as a Wellcome Trust Clinical Research Career Development Fellow (224586/Z/21/Z) and NIHR Moorfields Biomedical

Research Centre. MRC and the Clatworthy lab (NR) are supported by a Wellcome Investigator Award (220268/Z/20/Z), the National Institute of Health Research (NIHR) Cambridge Biomedical Research Centre (NIHR203312), and the NIHR Blood and Transplant Research Unit in Organ Donation and Transplantation (NIHR203332), a partnership between NHS Blood and Transplant, the University of Cambridge and Newcastle University. The views expressed are those of the authors and not necessarily those of the NIHR or the Department of Health and Social Care. F.C. and S.F. acknowledge funding support by the Federal Ministry of Education and Research (BMBF), as part of the National Research Initiatives for Mass Spectrometry in Systems Medicine, under grant agreement No. 161L0222. JC, SD, VMO and JFEK are supported by a peer-reviewed Food Allergy Research Grant jointly funded by the CIHR Institute of Infection and Immunity (CIHR-III), CIHR Institute of Circulatory and Respiratory Health (CIHR-ICRH), and the Canadian Allergy, Asthma and Immunology Foundation (CAAIF) and by the Walter and Maria Schroeder Foundation and J.P. Bickell Foundation. WOD is supported by CNPq, INCT-DT and FAPEMIG. SF is supported by NIH grant NIH R01AR077019. MYG is supported by NIH grant R01AI134713 and NIH contract 75N93019C00070. AG is funded by Damon Runyon Cancer Research Foundation (National Mah Jongg League Fellowship (DRG 2409–20)). KJG and KLPM are supported by CNPq, FAPEMIG, FAPESP (#2021/00408–6), Instituto Nacional de Ciencia e Tecnologia em Doenças Tropicais (INCT-DT). DJ is supported by the grant of the European Research Council (ERC); European Consolidator Grant, XHale (Reference #771883). WK is supported by the European Research Council (ERC) (819329-STEP2). AK and RBM are supported by a Biotechnology and Biological Sciences Research Council (BBSRC) David Phillips fellowship (BB/S010386/1). AYK is supported by the Wellcome Trust (222096/Z/20/Z). All research at GOSH is supported by the National Institute of Health Research (NIHR) GOSH Biomedical Research Centre (BRC). JCSL is supported by the São Paulo Research Foundation (FAPESP fellowship 2023/01697-7). VIM is supported by the NIGMS MOSAIC K99/R00 4R00GM147841-02. JMOG is supported by the National Polytechnic Institute (IPN) of Mexico (ID: DRI/DII/0445/2024) and the National Council of Humanities, Science and Technology (Conahcyt) (ID: B210344 and CVU: 10008) with fundings for research at UCL-Great Ormond Street Institute Child Health (UCL-GOSH). MV was supported by the Canadian Institutes of Health Research (CIHR) Doctoral Award (Grant No. 170793) and the Ontario Graduate Scholarship (OGS) Program.

## Additional information

### Competing interests

Joshua Croteau: JC is an employee and stakeholder of BioLegend (revvity inc). The author has no other competing interests to declare. Sarah A Teichmann: SAT is a remunerated Scientific Advisory Board member of BioOptimus, Foresite Labs, Xaira Therapeutics, and a consultant and equity holder of TransitionBio and EnsoCell, and a non-executive board director of 10x Genomics, as well as part-time employee of GlaxoSmithKline. Andrea J Radtke: AJR is an employee of Leica Microsystems. The author has no other competing interests to declare. The other authors declare that no competing interests exist.

### Funding

| Funder | Grant reference number | Author |
| --- | --- | --- |
| National Institutes of Health | HHSN316201300006W/75N93022F00001 | Ziv Yaniv |
| National Institutes of Health | OT2OD033756 | Katy Börner |
| National Institutes of Health | U24CA268108 | Katy Börner |
| National Institutes of Health | U24DK135157 | Katy Börner |
| National Institutes of Health | U2CDK114886 | Katy Börner |

| Funder | Grant reference number | Author |
|---|---|---|
| Wellcome Trust | 10.35802/224586 | Colin J Chu |
| Wellcome Trust | 10.35802/220268 | Menna R Clatworthy Nathan Richoz |
| National Institutes of Health | R01AR077019 | Spencer Fullam |
| National Institutes of Health | R01AI134713 | Michael Y Gerner |
| National Institutes of Health | 75N93019C00070 | Michael Y Gerner |
| Damon Runyon Cancer Research Foundation | DRG 2409-20 | Anita Gola |
| European Research Council | 771883 | Danny Jonigk |
| European Research Council | 819329-STEP2 | Wolfgang Kastenmüller |
| Biotechnology and Biological Sciences Research Council | BB/S010386/1 | Aanandita Kothurkar Ryan B MacDonald |
| Wellcome Trust | 222096/Z/20/Z | Alexandra Y Kreins |
| National Institutes of Health | K99/R00 4R00GM147841-02 | Vivien I Maltez |
| Canadian Institutes of Health Research | 170793 | Megan Vierhout |
| NIHR Cambridge Biomedical Research Centre | NIHR203312 | Menna R Clatworthy |
| NIHR Blood and Transplant Research Unit in Organ Donation and Transplantation | NIHR203332 | Menna R Clatworthy |
| CNPq, FAPEMIG, FAPESP | #2021/00408–6 | Kenneth J Gollob Katia Luciano Pereira Morais |
| São Paulo Research Foundation | FAPESP fellowship 2023/01697-7 | Jean CS Luz |
| National Polytechnic Institute (IPN) of Mexico | DRI/DII/0445/2024 | Juan M Ocampo-Godinez |
| National Council of Humanities, Science and Technology (Conahcyt) | B210344 | Juan M Ocampo-Godinez |
| National Council of Humanities, Science and Technology (Conahcyt) | CVU: 10008 | Juan M Ocampo-Godinez |

The funders had no role in study design, data collection and interpretation, or the decision to submit the work for publication. For the purpose of Open Access, the authors have applied a CC BY public copyright license to any Author Accepted Manuscript version arising from this submission.

## Author contributions
Ziv Yaniv, Conceptualization, Software, Supervision, Methodology, Writing – original draft, Writing – review and editing; Ifeanyichukwu U Anidi, Katy Börner, Colin J Chu, Menna R Clatworthy, Joshua Croteau, Walderez O Dutra, Michael Y Gerner, Kenneth J Gollob, Jonathan M Hernandez, Danny Jonigk, Wolfgang Kastenmüller, Joshua FE Koenig, John E McDonough, Annette Oxenius, Sarah A Teichmann, Ronald N Germain, Resources, Funding acquisition, Writing – review and editing; Leanne Arakkal, Armando J Arroyo-Mejías, Rebecca T Beuschel, Beatrice H Clark, Jake Colautti, Fabian Coscia, Saven Denha, Rose Dever, Sonja Fritzsche, Spencer Fullam, Anita Gola, Jyh Liang Hor, Hiroshi Ichise, Zhixin Jing, Evelyn Kandov, Aanandita Kothurkar, Rosa K Kortekaas, Alexandra Y Kreins, Ian T Lamborn, Yuri Lin, Katia Luciano Pereira Morais, Aleksandra Lunich, Jean CS Luz, Ryan B MacDonald, Chen Makranz, Vivien I Maltez, Ryan V Moriarty, Juan M Ocampo-Godinez, Vitoria Murakami Olyntho, Kartika Padhan, Kirsten Remmert, Nathan Richoz, Edward C Schrom, Wanjing Shang, Lihong Shi, Rochelle M Shih, Emily Speranza, Salome Stierli, Tibor Z Verse, Megan Vierhout, Brianna T Wachter, Adam K Wade-Vallance, Margaret Williams, Nathan Zangger, Data curation, Writing – review and editing; Andrea J Radtke, Conceptualization, Data curation, Supervision, Methodology, Writing – original draft, Project administration, Writing – review and editing

## Author ORCIDs
Ziv Yaniv ⓘ https://orcid.org/0000-0003-0315-7727
Ifeanyichukwu U Anidi ⓘ https://orcid.org/0000-0002-2692-1225
Colin J Chu ⓘ https://orcid.org/0000-0003-2088-8310
Menna R Clatworthy ⓘ https://orcid.org/0000-0002-3340-9828
Fabian Coscia ⓘ https://orcid.org/0000-0002-2244-5081
Saven Denha ⓘ https://orcid.org/0009-0004-8162-409X
Sonja Fritzsche ⓘ https://orcid.org/0000-0003-3335-3534
Spencer Fullam ⓘ https://orcid.org/0000-0001-9012-3802
Michael Y Gerner ⓘ https://orcid.org/0000-0001-5406-8308
Jyh Liang Hor ⓘ https://orcid.org/0000-0001-6721-4204
Hiroshi Ichise ⓘ https://orcid.org/0000-0002-5187-810X
Zhixin Jing ⓘ https://orcid.org/0000-0003-1118-7432
Joshua FE Koenig ⓘ https://orcid.org/0000-0002-8909-5039
Rosa K Kortekaas ⓘ https://orcid.org/0000-0002-8728-1735
Alexandra Y Kreins ⓘ https://orcid.org/0000-0001-8748-5837
Yuri Lin ⓘ https://orcid.org/0009-0006-9784-2694
Ryan B MacDonald ⓘ https://orcid.org/0000-0003-4194-8925
John E McDonough ⓘ https://orcid.org/0000-0003-2497-0258
Vitoria Murakami Olyntho ⓘ https://orcid.org/0000-0003-2634-6838
Wanjing Shang ⓘ https://orcid.org/0009-0007-0646-7946
Emily Speranza ⓘ https://orcid.org/0000-0003-0666-4804
Sarah A Teichmann ⓘ https://orcid.org/0000-0002-6294-6366
Megan Vierhout ⓘ https://orcid.org/0000-0001-5813-2581
Adam K Wade-Vallance ⓘ https://orcid.org/0000-0002-6863-1461
Nathan Zangger ⓘ https://orcid.org/0000-0003-1130-1899
Ronald N Germain ⓘ https://orcid.org/0000-0003-1495-9143

Reviewer #1 (Public review): https://doi.org/10.7554/eLife.105737.3.sa1
Reviewer #2 (Public review): https://doi.org/10.7554/eLife.105737.3.sa2
Reviewer #3 (Public review): https://doi.org/10.7554/eLife.105737.3.sa3
Author response https://doi.org/10.7554/eLife.105737.3.sa4

---

# Additional files

## Supplementary files
MDAR checklist

Supplementary file 1. Transcript of conversation with AI agent using GitHub Copilot and the Knowledge-Base as context.

## Data availability

This manuscript describes a publicly available resource/dataset. The most current version of the resource is available from https://github.com/IBEXImagingCommunity/ibex_imaging_knowledge_base (*Yaniv, 2026*). Authoritative versions are regularly released on the Zenodo repository: https://doi.org/10.5281/zenodo.7693278.

The following dataset was generated:

| Author(s) | Year | Dataset title | Dataset URL | Database and Identifier |
|---|---|---|---|---|
| Yaniv Z, Anidi IU, Arakkal L, Arroyo-Mejias AJ, Beuschel RT, Borner K, Chu CJ, Clark BH, Clatworthy MR, Colautti J, Coscia F, Croteau J, Denha S, Dever R, Dutra WO, Fritzsche S, Fullam S, Gerner MY, Gola A, Gollob KJ, Hernandez JM, Hor JL, Ichise H, Jing Z, Jonigk D, Kandov E, Kastenmuller W, Koenig JFE, Kortekaas RK, Kothurkar A, Kreins AY, Lamborn IT, Lin Y, Luciano Pereira Morais K, Lunich A, MacDonald RB, Makranz C, Moriarty RV, Ocampo-Godinez JM, Olyntho VM, Oxenius A, Padhan K, Remmert K, Richoz N, Schrom EC, Shang W, Shi L, Shih RM, Speranza E, Stierli S, Teichmann SA, Veres TZ, Vierhout M, Wachter BT, Wade-Vallance AK, Williams M, Zangger N, Germain RN, Radtke AJ, Maltez VI, McDonough JE, Luz J | 2025 | Iterative Bleaching Extends Multiplexity (IBEX) Knowledge-Base | https://doi.org/10.5281/zenodo.7693278 | Zenodo, 10.5281/zenodo.7693278 |

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
