## [Editor Report · eLife Assessment]

The IBEX Knowledge-Base is a **fundamental** tool that will enhance scientific collaboration by providing a centralized, community-driven resource for immunofluorescence imaging and reagent validation. Its detailed use cases, open-source design, and transparent reporting offer **exceptional** evidence of its broad utility and impact in the life sciences. It is now up to the community to contribute to its growth. Overall, the resource sets a high standard as a blueprint for future community initiatives in reproducibility and standardization.

---

## [Referee Report · Reviewer #1 (Public review)]

IBEX Knowledge Database

Here, Yanid Z. and colleagues present the IBEX knowledge base. A community tool developed to centralize knowledge and help its adoption by more users. Authors have done a fantastic job, and there is careful consideration of the many aspects of the data management and FAIR principles. The manuscript needs no further work, as it is very well written and have detailed descriptions for data contribution as well as describing the KB itself. Overall, it is a great initiative, especially the aim to inform about negative data and non-recommended reagents, which will positively affect the user community and scientific reproducibility.

This initiative will serve as a groundwork to include technical details of other multiple immunofluoresecence methods (such as immunoSABER, 4i, etc). Including other methods would help the knowledge base itself and related methods to evolve and assist their communities in the future.

Significant care has been taken to allow the report of negative data. While there might be limitations as to how this information is included, transparency and community usage will ensure the knowledge base offers a fair representation.

There are two ways to contribute to the knowledge base. While authors have contributed significantly to its creation, it will be the role of the maintainers to assist potential users and contributors. It is specially appreciated that a path to contribute is possible with no coding skills. I am keen to see how the KB evolves and it helps disseminate the use of this and other great techniques.

---

## [Referee Report · Reviewer #2 (Public review)]

Summary:

The paper introduces the IBEX Knowledge-Base (KB), a shared online resource designed to help scientists working with immunofluorescence imaging. It acts as a central hub where researchers can find and share information about reagents, protocols, and imaging methods. The KB is not static like traditional publications; instead, it evolves as researchers contribute new findings and refinements. A key highlight is that it includes results of both successful and unsuccessful experiments, helping scientists avoid repeating failed experiments and saving time and resources. The platform is built on open-access tools ensuring that the information remains available to everyone. Overall, the KB aims to collaboratively accelerate research, improve reproducibility, and reduce wasted effort in imaging experiments.

Strengths:

(1) The IBEX KB is built entirely on open-source tools, ensuring accessibility and long-term sustainability. This approach aligns with FAIR data principles and ensures that the KB remains adaptable to future advancements.

(2) The KB also follows strict data organization standards, ensuring that all information about reagents and protocols is clearly documented and easy to find with little ambiguity.

(3) The KB allows scientists to report both positive and negative results, reducing duplication of effort and speeds up the research process.

(4) The KB is helpful for all researchers, but even more so for scientists in resource-limited settings. It provides guidance on finding affordable alternatives to expensive or discontinued reagents, making it easier for researchers with fewer resources to perform high-quality experiments.

(5) The KB includes a community discussion forum where scientists can ask for advice, share troubleshooting tips, and collaborate with others facing similar challenges.

(6) The authors discuss plans for active maintenance of the database and also to incentivize higher participation from the community.

(7) Even those unfamiliar with Github may contribute with the help of the database maintenance team.

Note: The authors have addressed my comments on the previous version of the article and the current version has been strengthened as a result.

---

## [Referee Report · Reviewer #3 (Public review)]

Summary:

The authors have developed and interactive knowledge-base that uses crowdsourcing information on antibodies and reagents for immunofluorescence imaging.

Strengths:

The authors provide an extremely relevant and needed interphase for collaboration through a well-built platform. All the links in their website work, the information provided, reagents, datasets, videos and protocols are very informative. The instructions for the community researchers to contribute is clear and they provide detailed instructions in how to technically proceed. Additionally, the interface has been refined to enable the contribution regardless of the computational expertise of the researcher.

Weaknesses:

The Knowledge-Base relies on community contributions without mandatory, standardized metadata and validation criteria. Whilst this enhances the contributions, it limits the reliability of the database.

---

## [Author Response]

The following is the authors’ response to the original reviews

**Reviewer #1:**
(1) As such amount of work has been put into developing this community tool, it would be worth thinking about how it could serve other multiplex-immunofluorescence methods (such as immunoSABER, 4i, etc). Adding an extra tab where the particular method that uses those reagents is mentioned. This would also help as IBEX itself and related methods evolve in the future.

We agree and currently support six other methods beyond the original ”IBEX2D Manual”, with the most generic being ”Multiplexed 2D Imaging”: standard, single cycle (non-iterative) imaging method applied to thin, 2D (5-30 micron) tissue sections. Descriptions of supported methods are given in the reagent glossary. We plan to evolve to include multiplex IF methods such as Immuno-SABER, 4i, Cell DIVE, etc. The current structure of the reagent resources table can support other immunofluorescence methods without modifications. The table contains information for IBEX and related methods. The particular method for which a reagent validation was evaluated is specified in the column titled ”Method”. Descriptions of supported methods are given in the reagent glossary.

(2) It has a rather minimal description of the software. In particular, there is software that has not been developed for IBEX specifically but that could be used for IBEX datasets (ASHLAR, WSIReg, VALIS, WARPY, and QuPath, etc). It would be nice if there was mention of those.

ASHLAR, WSIReg, VALIS, and Warpy have been added to the Knowledge-Base. These software components are specifically relevant for iterative imaging protocols which require image alignment. With respect to QuPath, Fiji, Napari and other general microscopy image analysis frameworks, these are not listed. Such frameworks provide a wide range of operations relevant for many microscopy image analysis tasks and are likely already familiar to researchers who are interested in the information contained in the Knowledge-Base.

(3) There is a concern about how the negative data information will be added, as no publication or peer-review process can back it up. Perhaps the particular conditions of the experiment should be very well described to allow future users to assess the validity.

We agree with this observation and have added the following language to the contribute page:

”When reporting information that has not appeared in a peer-reviewed publication, both negative and positive results, include more details with respect to experimental conditions and provide sample images as part of the supporting material files. In all cases, peer reviewed or not, we encourage providing additional details in the supporting material that you deem important and are not part of the csv file structure. These include, but are not limited to, lot numbers, versioned protocols used in the work, and any other information which will facilitate validation reproducibility.”

(4) The proposed scheme where a reagent can be validated or recommended against by up to 4 different labs should be good. It may be good to make sure that researchers who validate belong to different labs and are not only different ORCID that belong to the same group. Similar to making a case of recommendations against a reagent.

We generally support this recommendation. Based on our experience, even members within the same laboratory encounter challenges when attempting to validate reagents contributed by current or former colleagues. Additionally, research labs often experience significant personnel turnover, with minimal overlap over a five year span.

To address these concerns, we have updated the instructions on the contribute page as follows: ”We only accept up to 5 ORCID additions in the Agree or Disagree columns. This means that the original contributor’s work was replicated by up to 4 individuals or refuted by up to 5 people. Priority is given to contributions from individuals in laboratories distinct from the original source.”

(5) It is very interesting to keep track of the protocol versions used. Perhaps users should be able to validate independent versions and it will be important to know how information is kept.

Thank you for your suggestion. We encourage members of the community to cite the latest version of the Knowledge-Base in the “Citing the Knowledge-Base” section.

(6) The final point I would make is that the need to form a GitHub repository may deter some people from submitting data. For sporadic contributions, authors could think that users could either reach out to main developers and/or provide a submission form that can help less experienced users of command-line and GitHub programming, but still promote the contribution from the community.

We have given this significant thought and now support a secondary path for contributing that does not require familiarity with git or GitHub. This path involves downloading a zip file, modifying the contents of the csv files and providing supporting material text files and images. Once the work is completed, the contributor contacts the Knowledge-Base maintainers and we complete the submission together, with the maintainers dealing with the usage of git and GitHub. This information has been added to the notes which are listed at the top of the Contribute page. We have recently completed the first contribution that followed this new workflow.

We still encourage researchers to familiarize themselves with git and the GitHub repository hosting service. These tools have been shown to be useful for collaborative and reproducible laboratory research.

**Reviewer #2:**
(1) The potential impact of IBEX KB is very clear. However, the paper would benefit by also discussing more on KB maintenance and outreach, and how higher participation could be incentivized.

We have added the following details to the discussion:

The KB is actively maintained by its chairs, who meet bi-weekly to ensure its continued development and maintenance. In addition to these regular meetings, we engage with both current and prospective community members to gather feedback, encourage contributions, and expand the collective knowledge supporting the KB. To broaden outreach and foster sustained engagement, the IBEX community will collaborate with synergistic initiatives such as the HuBMAP Affinity Reagents Working Group, the European Society for Spatial Biology (ESSB), and the Global Alliance for Spatial Technologies (GESTALT).

As a further incentive for participation, we intend to launch an annual “Reagent Validation Week”, a community driven event inspired by software hackathons. During this dedicated week, researchers would focus on validating or reproducing validation for selected reagents and contribute their findings to the KB. We have also discussed hosting an “Around the World” symposium, featuring presentations from both junior and senior scientists across the community, to showcase diverse perspectives and foster global collaboration.

(2) Use of resources like GitHub may limit engagement from non-coding members of the scientific community. Will there be alternative options like a user-friendly web interface to contribute more easily?

We agree with this observation and have addressed it. Please see detailed response to point 6 from Reviewer 1.

**Reviewer #3:**
(1) IBEX is a specific immunofluorescence method. However, the utility of the Knowledge base is not limited to the specific IBEX method. Therefore, I suggest removing the unnecessary branding of the term IBEX from the KB and citing potentially other similar cyclic immunofluorescence methods in the manuscript (e.g. CycIF Lin et al 2018). This would also emphasize the wider impact and applicability of the KB to the wider imaging community.

For now, we have decided to keep the original reference to the IBEX method in the resource name and re-brand it in the next development phase. In that phase we intend to solicit reagent validations for methods unrelated to IBEX. We have added the reference to the CycIF publication. The manuscript text now reads: “We are optimistic that future versions will include extension of the IBEX method to other tissues and species and we intend to solicit contributions of reagent validations for other multiplexed imaging techniques such as CycIF Lin et al. (2015). At that point in time we expect to re-brand the KB as the IBEX++ Knowledge-Base...”

(2) I believe reporting negative results with reagents is highly valuable. However, the way to report antibodies must include more details. To ensure data quality, every report should be linked to a specific protocol + images or doc with the standard document variations, and sample information. This should be a mandatory requirement.

We agree that this information is desirable, but we do not agree that it should be mandatory. In the contribution instructions we now explicitly list lot numbers and versioned protocols as examples of details that we encourage contributors to include in their supporting material files. We believe that requiring this information for a contribution sets the bar too high and will deter many from contributing information that can benefit others.

(3) While cross-validation among researchers is beneficial, even if five individuals fail to reproduce results with a given antibody, their findings may be influenced by techniquespecific factors. It is also important to consider whether these researchers come from the same group, institution, or geographical region, as this could impact reproducibility. Additionally, entries that have not been reproduced at least five times using the same protocol should still be considered valuable information. To address this, an ”insufficient validation data” flag could be implemented, ensuring that incomplete but useful findings remain accessible.

The contribution instructions now state that ”Priority is given to contributions from individuals in laboratories distinct from the original source”.

While our goal is to support reproducing reagent validations, we do not expect these type of contributions be the rule as the only incentive we can provide to encourage this behavior is co-authorship on the authoritative dataset. As a result, it is likely that many of the validations will have a single endorser, the original contributor. These results are valuable information and we do not think they should be singled out (insufficient validation label). We leave it up to the users of the KB to decide whether they trust recommendations with multiple endorsers or if endorsement by a single highly trusted contributor is sufficient for them. In all cases, issues with contributions can be rasied and discussed on the KB discussion forum.

The rationale for limiting the number of reproduction studies to five was that this is a minimal, yet sufficiently large, number that provides confidence in the results. Placing an upper limit ensures that researchers do not provide reproduction results for widely used and well established reagents just because these results are readily available to them.

(4) This system could flag reagents with inconsistent reports, highlight potential techniquespecific issues, and suggest alternative reagents with stronger validation records. Furthermore, a validation confidence ranking could be introduced, taking into account the number of independent confirmations, protocol consistency, and reproducibility data. These measures would help refine the reporting process while maintaining transparency and scientific rigor.

We agree that the functionality described here is desirable, but this is not part of the KB. At its core the KB is a dataset and we do not envision developing dedicated tools to perform these tasks. Instead, we foresee using the KB as context for interacting with AI agents. Providing the KB as context to an AI, one can currently use it to answer domain specific questions and perform related tasks such as designing imaging panels (under subject matter expert supervision). This was added to the sample usecases in the manuscript with a transcript from interaction with an AI model using the website as context provided as supplemental material.

(5) Regarding image formats for results reporting, while JPG files are convenient due to their small size, TIFF files offer significant advantages, such as preserving metadata and maintaining the integrity of real data values. Proper signal adjustments may not always be applied by researchers, making TIFF crucial for accurate data analysis. I suggest in this regard making available the possibility of including a link to the original TIFF data

The goal of the supporting material image is similar to that of an image used in a manuscript and it should not be used for data analysis purposes. This is the reason we chose the JPG format. Sharing these images is not intended to be a substitute for publicly sharing the original images and their associated metadata. This is now noted in the contributing instructions.

(6) Homepage:Include a brief summary of the knowledge base’s purpose and tabs to provide clarity for new users. The current homepage is a bit misleading for newcomers.

The homepage has been modified to include information about the Knowledge-Base, contents and how to use it including as context for interaction with AI agents.

(7) Reagent Resources Section:Enable users to search for a target name directly, rather than filtering through dropdown options.

The dropdown menu explicitly shows all available targets and also allows for direct search of target name. To use it for direct search, once the dropdown is selected start typing the name of the target and the focus will jump to it. Thus, if looking for ”Zrf1” there is no need to scroll through all targets in the dropdown. This also facilitates easy clearing of a filter, select the dropdown and start typing the word ”clear”, then press enter when it is highlighted. This information has been added to the page.

Provide an option to download the dataset as a CSV file. This feature will be highly valued by non-computational researchers.

Links to download the reagent resources csv file and the whole Knowledge-Base have been added.

Add the same column documentation here as in the contributor instructions. For example, you need to make clear the distinctions between ”Recommend,” ”Agree,” and ”Disagree” ratings, as they may be misleading to those who have not visited the rules to contribute.

A link to the column documentation in the contributor instructions has been added here. Information on the website is displayed in one location and linked as needed. Duplicated display of information creates uncertainty for users and results in more complex instructions when referring to the information.

Include additional details in the dataset, such as lot numbers, or the date of the contribution, that could be relevant in different settings.

Please see response to point 2.

(8) Data & Software Section:Add filtering options in the table based on organism and tissue availability

This data is not encoded in the available information in an independent manner so we do not directly enable filtering. It is usually included in the ”Details” free form text. This text is duplicated from the original dataset descriptions. One can still search this page using the browsers search functionality to achieve behavior similar to filtering. While the ”Details” text may not be visible due to the usage of the accordion user interface, it is still searchable and will automatically expand when the search text is found under the collapsed accordion button.

(9) Contributor Section:Incorporate figures from the manuscript to make it more visual and improve understanding of rules and standards.

Figure 4 from the manuscript was added to this page.

I believe reporting negative results with reagents is highly valuable. However, to ensure data quality, every report should be linked to a specific protocol and sample information. This should be a mandatory requirement. To streamline the process, warnings for certain reagents could be implemented, but a reagent should not be outright labeled as ineffective without proper validation.

Please see response to point 2.

Cross-validation among researchers is beneficial, but even if five individuals fail to reproduce results with a given antibody, it may still be due to technique-specific factorsparticularly for non-routine antibodies.

We agree with this observation and have modified the contribution instructions accordingly:

When overturning previously reported results, the number of ORCIDs in the Disagree column becomes greater than those in the Agree column, we will open the contribution for public discussion on the Knowledge-Base forum before accepting it.

The intent is to increase the community’s confidence in the results, particularly when dealing with non-routine antibodies. This allows the original contributor and other members of the community to engage with the researchers who were unable to replicate a specific validation, possibly helping them to replicate the original results by adding missing details to the KB, or explicitly identifying and documenting issues with the original work.

Regarding image formats, JPG files are convenient due to their small size, but TIFF offers significant advantages, such as preserving metadata and maintaining the integrity of real data values. Proper signal adjustments may not always be applied by researchers, making TIFF crucial for accurate data analysis.

Please see response to point 5.